# Structural Analysis of Janus Tyrosine Kinase Variants in Hematological Malignancies: Implications for Drug Development and Opportunities for Novel Therapeutic Strategies

**DOI:** 10.3390/ijms241914573

**Published:** 2023-09-26

**Authors:** Omar J. Rodriguez Moncivais, Stephanie A. Chavez, Victor H. Estrada Jimenez, Shengjie Sun, Lin Li, Robert A. Kirken, Georgialina Rodriguez

**Affiliations:** 1Department of Biological Sciences, The University of Texas at El Paso, 500 W. University Ave., El Paso, TX 79902, USA; 2Border Biomedical Research Center, The University of Texas at El Paso, 500 W. University Ave., El Paso, TX 79902, USA; 3Department of Physics, The University of Texas at El Paso, 500 W. University Ave., El Paso, TX 79902, USA; 4Computational Sciences Program, The University of Texas at El Paso, 500 W. University Ave., El Paso, TX 79902, USA

**Keywords:** hematological malignancies, Janus tyrosine kinase, gain-of-function mutations, computational modeling, homology modeling

## Abstract

Janus tyrosine kinase (JAK) variants are known drivers for hematological disorders. With the full-length structure of mouse JAK1 being recently resolved, new observations on the localization of variants within closed, open, and dimerized JAK structures are possible. Full-length homology models of human wild-type JAK family members were developed using the Glassman et al. reported mouse JAK1 containing the V658F structure as a template. Many mutational sites related to proliferative hematological disorders reside in the JH2 pseudokinase domains facing the region important in dimerization of JAKs in both closed and open states. More than half of all JAK gain of function (GoF) variants are changes in polarity, while only 1.2% are associated with a change in charge. Within a JAK1-JAK3 homodimer model, *IFNLR1* (PDB ID7T6F) and the IL-2 common gamma chain subunit (IL2Rγc) were aligned with the respective dimer implementing SWISS-MODEL coupled with ChimeraX. JAK3 variants were observed to encircle the catalytic site of the kinase domain, while mutations in the pseudokinase domain align along the JAK-JAK dimerization axis. FERM domains of JAK1 and JAK3 are identified as a hot spot for hematologic malignancies. Herein, we propose new allosteric surfaces for targeting hyperactive JAK dimers.

## 1. Introduction

### 1.1. Molecular Structure of Janus Tyrosine Kinase (JAK)

The JAK family of JAK1, JAK2, JAK3, and TYK2 are critical drivers of hematological development, differentiation, proliferation, and survival by over 40 cytokine [1] and growth factor receptors [2,3,4]. JAKs are classified as a family based on their shared seven JAK homology (JH) domains numbered sequentially from the carboxyl to the amino terminal end (Figure 1A). Sequence similarity (>40%) across seven Janus homology (JH) domains and four functional domains, enzymatic activity, shared targets, and response to cytokines defines these tyrosine kinases as a family. At the amino-terminal (N-terminal) end of JAKs (JH6–JH7) spans an erythrocyte band 4.1, ezrin, radixin, moesin (FERM) domain (Figure 1A) [5] involved in plasma membrane localization and association with cytokine receptors [6,7]. The adjacent Src-homology 2 (SH2) domain encompasses JH3-JH4 [5] of JAKs and controls protein–protein interactions including receptor association via tyrosine phosphorylated residues [8]. Carboxyl-terminal (C-terminal) JH domains harbor an inactive pseudokinase (JH2) and catalytically active kinase domain (JH1) [5]. Segments between functional domains form connective linkers between FERM-SH2 (linker of 9 to 20 amino acids), SH2-JH2 (linker of 40 to 60 amino acids), and JH2-JH1 (linker of 20 to 40 amino acids). Appendix A contains details of individual JAK regions.

Tandem pseudokinase and kinase domains within JAKs share conserved N-lobe, C-lobe, and secondary structural motifs (Figure 1B). The central catalytic cleft between the N-lobe and C-lobe is the area where nucleotide (ATP/ADP) binds and phosphotransferase activity occurs. Secondary structures and motifs are highly conserved between JH2 and JH1 domains. 

### 1.2. Signal Transduction and Phosphorylation of Janus Tyrosine Kinases

JAKs associate with Type I (common gamma chain family, common beta chain family, gp130 family, GH, Prl, Epo, TPO, G-CSF, leptin) and Type II (IFN family, IL-10, IL-20) receptors which lack intrinsic tyrosine kinase activity for signal transduction from the extracellular to intracellular space [1,9]. The extracellular ligand binds to homodimeric, heterodimeric, or heterotrimeric receptors, causing aggregation of JAKs which associate with intracellular proline rich and hydrophobic box motifs of the receptor [10,11,12]. JAK dimerization allows for autoactivation through transphosphorylation of tyrosine residues within the activation loop of each JH1 kinase domain. JAK and Signal Transducer and Activator of Transcription (STAT) proteins are primarily regulated by phosphorylation events. Phosphorylation of Y1034 within JAK1 [13], JAK2 Y1007 [14,15], JAK3 Y980 [16], and Y1054 [17] of TYK2 allows for the activation loop to open and promotes phospho-transferase of the gamma-phosphate of ATP to substrate tyrosine residues. Alternatively, phosphorylation of tyrosine residues within the activation loop, i.e., JAK3 Y981, is associated with termination of JAK activity [16]. For a comprehensive list of known JAK post-translational sites, refer to Appendix A. Activated JAKs are capable of phosphorylating cytokine receptors on immunoreceptor tyrosine-based activation motifs (ITAM) creating docking-sites for the SH2 domain containing proteins such as STATs [18]. STATs themselves become JAK substrates, allowing them to form hetero- or homodimers through their own SH2 domains. Some STAT proteins require serine phosphorylation for full activation and translocation to the nucleus [19] where they regulate transcription of genes related to anti-apoptotic and proliferative functions. Termination of the JAK/STAT signaling cascade involves dephosphorylation/phosphorylation, production of inhibitory proteins, and proteasomal degradation of receptors [20].

### 1.3. JAK Gain of Function Mutations

Gain of function (GoF) mutations within JAK proteins are associated with overproduction of immature hematological precursor cells leading to neoplasms, acute lymphoblastic leukemia (ALL), and acute myelogenous leukemia (AML). While genetic variants within the four JAK functional domains are associated with malignancies, the most well studied focus on the pseudokinase and kinase domains as these regions are considered “hot spots” for oncogenic mutations [21] GoF mutations within these regions are associated with hyperactive JAKs and JAK/STAT signaling pathways [22]. The most notable JH2 domain genetic alterations include JAK1 V658F and the homologous JAK2 V617F mutation known to cause Polycythemia vera (PV) [23,24]. The JAK pseudokinase homologs JAK2 R683G and JAK1 R724H appear to reside within an important area as more than one alteration of these residues is related to leukemia, R683S/G, and R724H/Q/S. JAK2 R683G has been highly associated with ALL cases related to down syndrome and one analysis showed that from 20 ALL children patients, 10 had this mutation. Comparably, JAK1 R724H has been related with B-ALL conferring IL-3 independence to growth and is found in T-ALL patients [25]. It is widely believed that mutations mapped to the FERM and SH2 domains join the receptor complex leading to JAK activation in the absence of cytokine or growth factors. Variants located within functional domain linkers are also capable of deleterious consequences. For example, JAK3 SH2-JH2 linker M511I, A573V, Q507P, and R657Q are transforming in nature [26]. JAK3 Q507P is associated with constitutive STAT5 activation and leads to T-cell prolymphocytic leukemia [27,28,29].

### 1.4. Open and Closed JAK Kinase Formations

Incomplete JAK crystal structures [30] have limited the understanding of intramolecular and intermolecular interactions within a full structure required for signal transduction. Much of what is known is based on the partial structures of FERM-SH2, pseudokinase, and kinase domains [30,31,32,33,34,35]. Recently, the full-length structure of mouse JAK1 V658F protein bound to partial Interferon lambda receptor 1 (*IFNLR1*) was established by cryo-EM [36]. Based on the established Glassman et al. structure, here we modeled the full-length open and closed human structure of JAK1, JAK2, JAK3, and TYK2 using SWISS-MODEL and Chimera-X. Furthermore, partial common gamma chain-containing Box1 and Box2 sequences were aligned within the JAK 3 model to predict receptor association. This study sought to understand the molecular-structural basis of gain-of-function (GoF) variants and hematopoietic neoplasms using the established full-length mouse JAK1 structure [36]. For an in-depth analysis of inflammatory disease and immunity GoF and loss-of-function (LoF), we recommend Ott N, et al. *JAKs and STATs from a Clinical Perspective: Loss-of-Function Mutations, Gain-of-Function Mutations, and Their Multidimensional Consequences*, J. Clin. Immunol., 2023 [37].

## 2. Results

### 2.1. Open Structure of JAK Kinases and Mutational Landscape in Hematological Malignancies

For decades, understanding the molecular mechanism of JAK mutations has been hindered by the lack of a full-length enzymatically active structure represented by the open structure bound to the cytokine receptor. The large size of JAK proteins (120–130 kDa) made previous determination of the complete JAK/receptor complex difficult. Here the wild-type (WT) full-length human JAK1, JAK2, JAK3 and TYK2 forms have been modeled using the Glassman et al. crystal structure of mouse JAK1 bound to the Interferon lambda receptor 1 (IFNLR1). ChimeraX and SWISS-MODEL were used to align individual JAK sequences with PDB 7T6F to develop the open forms of human JAK1 (Figure 2A), JAK2 (Figure 2B), JAK3 (Figure 2C), and TYK2 (Figure 2D). In this state, JAKs are observed to have accessible JH2 and JH1 domains (Figure 2) presumably to allow ATP and substrate interaction.

A mutational map of known hematological malignancy variants (partial summarized list, Table 1; for a full list see Appendix A) are depicted in Figure 2. Kinase domain sites are observed to group within the β1, β2, and β3 strands of JH1 and represent 15–30% of JAK variants (Figure 2E). Among the JAK family, JAK2 contains the most linker segment variants. The SH2 domain of all JAKs contains the least amount of reported hematopoietic malignancy mutations within functional domains (Figure 2E). Unsurprisingly, the majority of JAK transforming mutations reside within the JH2 pseudokinase domain among JAKs (Figure 2E). Notably, many JAK1 and JAK2 mutations cluster near the Glassman et al. JAK dimerization region of the JH2 domain (Figure 2A,B). A small portion (2.33%) of JAK2 and JAK3 mutations switch the amino acid charge, either from positive to negative or from negative to positive (Figure 3A). Among all combined Table 1 JAK variants, 44.6% did not change either polarity or charge, while 54.2% were associated with a change in polarity and 1.2% with a change in amino acid charge (Figure 3A). Within the JH2 domains of JAK1-3, more than half of variant changes are of polarity (Figure 3B).

### 2.2. JAK2 GoF Mutations Cluster to the JH1-JH2 Interface of the Closed Structure

Following cytokine-receptor complex formation, JAKs undergo structural changes from closed to open states to allow for the active site of the kinase domain to bind ATP and transfer gamma-phosphate to the substrate. We examined the process by which mutations could facilitate the enzyme open state in the absence of cytokine-receptor initiation. Among JAK family members, JAK2 is associated with the most hematopoietic malignancy variants (Appendix A). For this reason, the mutational landscape of JAK2 in the closed state is presented as a representative of JAK-driven hematopoietic neoplasms (Figure 4). Several mutations are observable in the region where the FERM domain, the SH2-JH2 linker, the pseudokinase domain, and the kinase domain converge. A mutational hot spot is evident in the region where the JH2 domain interacts with the JH1 domain. The SH2-JH2 linker runs through this space. In this closed state, the JAK dimerization region within the JH2 N-lobe forms an interface with the N-lobe of the JH1 kinase domain. The region spanning the JH2 domain (L545 to E627) β1, β2, β3, αC, β4, and β5, and JH1 (L849 to I951), β1-β6, αC, and αD may serve to negatively regulate the JH1 domain by binding directly and maintaining a closed state incapable of phospho-transferase activity. Interestingly, a region where JH1-JH2-FERM domains interact is observable in the closed JAK2 state.

### 2.3. Landscape of GoF Variants within the JAK1-JAK3 Dimer

A model of the JAK1-JAK3 dimer associated with the Type II cytokine receptor *IFNLR1* and Type I cytokine receptor common gamma chain (γc) for Interleukin –2 (IL-2), respectively, (Figure 5A) was created by using SWISS-MODEL and Chimera-X based on Glassman et al. (PDB ID 7T6F). Thirteen JAK1 mutations are found in highly dynamic loops, while twelve are in well-defined secondary structures. Comparably, nineteen JAK3 variants are found in loops and twelve are in well-defined secondary structures depicted in Figure 5A. Observably, three different regions of hematopoietic proliferative mutations are recognizable: (1) FERM, (2) pseudokinase, and (3) kinase.

Variants within the FERM domain of JAK1 and JAK3 are spread across the surface and do not immediately appear to cluster. Within the Interferon lambda receptor 1 (*IFNLR1*) and JAK1 (PDB ID 7T6F) structure, the JAK1 I62 and S71 sites of mutation contact the receptor (Figure 5B). Additionally, the SH2 sites T478 and S512 also contact *IFNLR1* near Box2 (Figure 3B). JAK1 S512 points directly toward Box2 of the *IFNLR1*. Presumably, these sites could impact interaction with the receptor. Similarly, alignment of the intracellular region of the IL2Rγc with JAK3 (Figure 5C) shows that the Box 1 motif of the receptor contacts JAK3 FERM residues P151 and L156, and these are proximal to R279 and Q283. JAK3 mutational sites G62, P132, and R172 of the FERM domain and SH2 R403 are near IL2Rγ Box 2. These residues lie within the same region of E183, which also does not make contact with the intracellular receptor.

As noted previously, the majority of JAK hyperactivating mutations are located in the pseudokinase domain (Figure 3). These sites lay in the same plane (Figure 5D), including some reported in the SH2-JH2 linker around the nucleotide binding motif (HRD/HRN) and the interaction interface between JAKs. This observation suggests that this region could be important in transactivation of JAKs, possibly by acting as a scaffold which permits activation of other proteins. Moreover, this could enhance the open structure and dimerization.

Lastly, JAK3 kinase domain variants encircle the enzyme active site (Figure 5E). Based on this observation, a possible mechanism of transformation may relate to ATP binding and stabilizing the phosphotransferase process.

An overlay of full-length human JAK1 and JAK3 shows a clustering of FERM-SH2 domain hematopoietic neoplasm variants along the surface pointing away from the center of the JAK dimer (Figure 6). This hot spot was not immediately observable above in Figure 5 for either independent JAK. Hot-spot residues are not near the predicted cytokine receptor alignment which lays at the center of the receptor-JAK dimer complex (Figure 5). The method of oncogenesis here is not apparent and illustrates the involvement of a multitude of mechanisms in hyperactivity.

### 2.4. PDBe Proteins, Interfaces, Structures, and Assemblies Analysis of the JAK1-JAK3 Dimer

Analysis of the heterodimer interfaces using the PDBe PISA tool (Proteins, Interfaces, Structures and Assemblies) was performed using default parameters (Figure 7). Two surface areas of interaction are noticeable within the PDBe PISA structure: (1) within the JAK FERM domain between JAK1 E355 and JAK3 L114 and (2) numerous JAK JH2 points. Twenty-four JAK1 and seventeen JAK3 amino acids are indicated to be dimer interface residues (Table 2). The number of residues predicted to be present within the interface regions constitute a small fraction, 1.6% and 2.1%, respectively, of the total monomer residues (Table 2). JAK1 interface residues include E355, M570, S571, Q572, L573, S574, F575, D576, R577, L580, D608, **R629, D630**, S632, L633, F636, E637, R643, V656, **V658**, R659, D660, V661, and N663. JAK3 interface residues include L114, L508, S509, Q510, **M511**, T512, F513, H514, L515, L563, N564, M566, E567, L570, E571, M592, and G594. JAK residues in bold font are known hematological malignancy variants. The following JAK1 interface residues are positioned proximal (<5 amino acids) to a reported hematological variant: E355 (**R360**), R629/D630 (**I631**), L633 (**A634**), and R643 (**S646**). Similarly, JAK3 residues include L508 (**Q507**), E571 (**A572/A573**), M592 (**A593**), and G594 (**A573**). Hematological variants in the latter and former lists are indicated by bold font.

In addition to information regarding accessible/buried protein surface areas and regions of interaction, the tool predicts the presence or absence of hydrogen bonds, salt bridges, and disulfide bonds (Table 2). The complexation significance score (CCS) determined under these conditions to be zero (CCS = 0) indicates that the interface interactions do not form a stable dimer complex. This is supported by the slight change observed in the complex solvation energy (Table 2). Within the dimerization interface, salt-bridges are predicted to form between JAK1 E637 and JAK3 L515 as well as R577 of JAK1 and JAK3 E571 (Table 2). Ten potential hydrogen bonds (<4 Å) are predicted to form between JAK1-JAK3 (Table 2).

## 3. Discussion

An abundance of variants within the JAK family are associated with hematological malignancies including T-ALL, B-ALL, ETP-ALL, AML, MPNs, PV, NKTCL, and AMKL (Table 1) and serve to emphasize their critical roles in immune development and homeostasis. With the establishment of a full-length JAK1 V658F (130 kDa) bound to partial *IFNLR1* it is possible to re-examine the location of hematological malignancy driving JAK variants. Here the open-state full-length human WT JAK models were created using the mouse Glassman et al. structure (Figure 2). Variants were observed to cluster within the JH2 N-lobe described to be the region of JAK dimerization. Notably, the majority of variants are found in this area across JAK1-3 (Figure 2). Conceivably, alterations in this region might promote dimerization and cytokine-independent signal transduction. TYK2 is associated with the fewest hematological neoplasm variants overall and is more commonly associated with inflammatory disorders.

SWISS-MODEL was also used in this study to create the JAK1-JAK3 heterodimer structure (Figure 5). An alternative approach would be to employ the Alphafold2 Multimer V3 tool [38] in combination with PDB 7T6F and template mode. Comparison of the WT JAK heterodimer using these two methods showed minor differences within loops and the position/orientation of variants between the modeled FERM and JH2 domains (Appendix A). The most drastic difference between the two models occurred with the orientation of the JAK3 kinase domain (Appendix A). This is unsurprising given that the PDB 7T6F template used with Alphafold2 is a JAK1 V658F homodimer. As a result, JAK3 in the Alphafold2 model looks to be in a transactivation position, while in the SWISS-MODEL structure (Figure 5), JAK3 more closely resembles JAK1 PDB 8EWY (Appendix A). Further experimental approaches are needed to draw solid conclusions.

Adopting the SWISS-MODEL structures, many of the JAK1 and JAK3 leukemia-driving variants are located within dynamic regions including small loops (Figure 2). More than a third of JAK mutations analyzed here are located in the pseudokinase domain (Figure 3). This area affects the interface with dimerized JAK molecules, as most of the mutations of JAK3 are in the pseudokinase domain. Hyperactivating JH2 mutations may maintain the JAK-JAK interface as opposed to blocking the active site or enhancing the interaction with the cytoplasmic domain of receptors. This idea is supported by the fact that pseudokinase domains can regulate partnering JAKs in trans as opposed to solely through cis regulation as has been reported for JAK1 GoF mutations [39,40].

The closed state of JAKs was modeled using the Glassman autoinhibited model of JAK1 [36] as a template. In this configuration, most variants lie between the JH2-JH1 interface where they likely destabilize the inhibitory interface as previously suggested [4,30]. Noticeably, a region where JH1-JH2-FERM domains interact is observable in the closed JAK state (Figure 4). In this formation, JAK2 JH2 N-lobe mutation sites I682 and R683 contact the JH1 N-lobe mutational residues D873 and T875. These interactions do not occur in the open state of JAK2. We hypothesize that mutations in this region help to keep the structure open, enhancing the process of dimerization and maintaining activated JAK-STAT signaling. Experimental studies are needed to support the observations made here. Past TYK2 pseudokinase and kinase domain modeling of the closed state [30] similarly showed the majority of reported leukemia mutations face the JH1-JH2 interface.

Like previous reports based on partial JAK structures [30,41,42,43,44], we observed many transforming mutations reside within the JH2 pseudokinase domain (Figure 3). Surprisingly, less than 3% of JAK1-3 variants are related to a change in side-chain charge and less than 50% of sites experience no change in polarity or charge, while greater than 50% of variants mentioned in Table 1 change polarity of the site (Figure 3A). An overwhelming number of JAK1-3 pseudokinase mutations are a change in polarity or involve no change in polarity/charge (Figure 3B). Of note, many of the JAK1 and JAK2 mutations cluster near the dimerization region of the Glassman et al. structure and nucleotide binding site. In fact, the binding of nucleotide to the catalytically inactive JH2 is involved in oncogenesis of JAK3 mutants [45]. These mutations are in highly dynamic structures such as loops that could be difficult to target with non-covalent or even covalent inhibitors since there is no observable characteristic pocket. This could explain why even when blocked the active site serves as a scaffold for other JAKs. Most mutations are present in the loop between the β3 and the αC of the pseudokinase domain. Since these regions are highly dynamic, minor changes could affect the interaction and transactivation process between partnering JAK homo- and heterodimers. Moreover, these areas may be more susceptible to post-translation modifications and/or mutations. Their presence within the JH2 β3 and αC loop suggests the importance of these secondary structures in controlling the JAK1 function. In fact, JAK1 JH2 mutations make contact with JAK3 once they are activated (Figure 5), suggesting that hyperactivating JH2 mutations maintain the JAK1-JAK3 dimer. Analysis of protein interactions, buried/accessible dimer surfaces, and complementary binding forces established by PDBe PISA revealed a small fraction of soluble surface residues within the dimerization interface of the JAK1-JAK3 complex (Figure 7, Table 2). A number of these residues are positioned near or are known hematological malignancy JAK variants. The PDBe PISA CCS = 0 calculation and slight change in solvation energy are not surprising given that the process of transactivation is highly dynamic and occurs rapidly. However, the blockade of this region with targeted inhibitors may be useful in disrupting ligand-independent dimerization and signal transduction.

Cumulatively, the placement of JAK mutations continues to support previous explanations of the mechanism of negative regulation by the pseudokinase domain [30,46]. Variants near the JH2 catalytic cleft potentially modulate kinase activity through the nucleotide-binding site, which has been shown to stabilize and modulate activity of JAK1, JAK2, and TYK2 [35,40,47].

In comparison, kinase domain mutations are fewer in number and are closely located to the kinase active site. JAK1 mutations R879S/C/H, T901G, and K908T reside within β1, a loop between β2 and β3, and β3, respectively. It has been established that the kinase-dead JH1 mutation K908A confers a catalytically inactive enzyme that reduces phosphorylation of downstream signaling factors including STATs [39,48]. Unexplainably, one study found that JAK1 K908A in combination with the JH2 mutation S703I rescues enzyme activity and increases STAT phosphorylation in the presence of JAK2 or TYK2 [39]. The kinase domain sites are observed to group within β1–β3 sheets which could be the area of substrate association implicating yet another mechanism that causes cytokine-independent signal transduction [49,50].

JAK FERM domains house the third highest group of mutational sites. Upon first examination, there is no observable clustering of mutations within individual JAKs (Figure 2); however, an overlay of JAK1 and JAK3 reveals that variants point outward away from intracellular portions of the cytokine receptor (Figure 6). It is widely believed that mutations mapped to the FERM and SH2 domains join the receptor complex leading to JAK activation in the absence of cytokine or growth factors; however, hematological malignancy mutations plotted here are not near the aligned cytokine receptors IFNR1 or IL-2Rγc. The FERM mutational surface (Figure 6) should be considered for allosteric drug targeting as this area can clearly regulate hyper-enzymatic activity.

Variants within linker segments indicate that these regions are capable of allosterically regulating enzymatic activity. JAK3, expressed primarily in hematopoietic cells, contains transforming mutations in the SH2-JH2 linker region (Q507P, M511I, M592V/T, A593V) and one mutation in the JH2-JH1 linker segment (V765D). M511I and A593V cause T-ALL and are sensitive to the JAK inhibitors tofacitinib and ruxolitinib [26,51].

Challenges and limitations of drug development against JAKs include the use of ATP-binding pocket competitive inhibitors, which often result in drug insensitivity and harsh side-effects during treatment. Alternative allosteric inhibitors which target regions distal from the active site represent practical options for obtaining JAK selectivity and avoiding secondary side-effects. The FDA has approved the use of allosteric JAK inhibitors for the treatment of immunological disorders such as SOTYKTU (Bristol-Meyers) for moderate to severe plaque psoriasis. Similarly, new therapeutic strategies should consider blocking the JH2 dimerization region to prevent hyperactive JAK transactivation events. For example, the JAK1 pseudokinase domain mutation V666G showed diminished in vitro activity and reduced activation of JAK3 by transactivation, suggesting that this area may be targetable for hyperactive JAK signaling. In contrast, the new JAK1 inhibitor VVD-118313 allosterically binds C817 of the pseudokinase domain. Residue C817 is distal from the JH2 JAK dimerization region. This treatment partially blocks IL-2 activated STAT5 phosphorylation in immune cells [52]. Because numerous JAK1 and JAK3 variants associated with hematological malignancies are located within dynamic structural loops (Figure 2), it is conceivable that these regions may be more susceptible to allosteric drug-targeting.

In 2009, the first JAK inhibitor (JAKi), ruxolitinib, was approved by the FDA for treatment of hematological malignancy myelofibrosis (MF). Since then, JAKis such as ruxolitinib, fedratinib, and pacritinib are FDA-approved drugs for the treatment of intermediate- to high-risk myelofibrosis (MF), a rare form of chronic leukemia [53,54,55]. Ruxolitinib is a JAK1/2 kinase ATP-competitive binding inhibitor associated with grade 3–4 anemia in 45.2% of recipients, thrombocytopenia in 12.9%, and neutropenia in 7.1% [56]. The most recent JAK inhibitor for the treatment of MF is pacritinib, a multikinase inhibitor effecting JAK2/FLT3/IRAK1/CSF1R. Disappointingly, the discussed JAKis are associated with adverse events including cyopenias, disease progression, and/or low therapeutic effects [57]. Clearly, a need exists for less toxic, more specific, small molecule inhibitors of JAK kinases. Ideally, therapeutic JAK inhibitors would target a mutated protein, leaving wild-type kinases in unrelated tissues unaffected. JAK inhibitor development should consider regions involved in cytokine receptor binding, JAK dimerization, and the primary mutation causing hyperactivity to personalize and provide meaningful treatment.

Limitations of this structural analysis include the lack of consideration for tyrosine phosphorylation and the impact of post-translation modifications on structure and function. Experimental studies are needed to support the observations made.

## 4. Materials and Methods

### 4.1. Human Full-Length JAK Modeling

UniProtKD sequences for human JAK1 (P23458), human JAK2 (O60674), human JAK3 (P52333), and human TYK2 (P29597) were used for full-length JAK modeling with SWISS-MODEL (Swiss Institute of Bioinformatics, Basel, Switzerland) [58,59,60,61] and UCSF ChimeraX (Resource for Biocomputing, Visualization, and Informatics, San Francisco, CA, USA) [62,63] as follows. Wild-type human JAK1 open and closed structures were developed by aligning the crystallographic structure of mouse JAK1 dimer with *IFNLR1* (PDB ID: 7T6F, [36]) and human TYK2 domains (PDB ID: 4OLI, [30]) as templates, respectively. SWISS-MODEL was used to diminish clashes and minimize the energy for the final structure of JAK1, and this model was used as a template to generate JAK2 open and closed structures and JAK3 and TYK2 open structures. The JAK3-JAK1 dimer was generated using as a template the mouse JAK1 dimer with *IFNLR1* (PDB ID: 7T6F, [36]). Since JAK3 interacts with IL2Rγ, the intracellular region containing Box1 and Box2 domains was generated by using SWISS-MODEL. The resulting structure was collocated using UCSF ChimeraX. Using MatchMaker resulted in a sequence alignment score of 35.1. Molecular analysis and localization of JAK sites were performed with UCSF ChimeraX.

### 4.2. PDBe PISA Tool

UniProt entries for JAK3 (P23458) and JAK1 (P52333) were used to analyze a WT-WT heterodimer with the PDBe PISA tool at https://www.ebi.ac.uk/pdbe/pisa/ (accessed on 18 August 2023) (EMBL’s European Bioinformatics Institute, Hinxton, Cambridgeshire, UK) [64]. Accessible and buried protein surface areas, the presence of hydrogen bonds, salt bridges, and disulfide bonds were obtained using default parameters: CRYST1 contains a = b = c = 1.0, alpha = beta = gamma = 90 degrees, space group = P 1. 

## 5. Conclusions

The landscape of JAK GoF variants associated with hematological malignancies was examined within the open and closed structures of JAK family members. Many GoF mutations analyzed were found in dynamic small loops and involve a change in polarity which could make inhibitor design difficult. JH2 domain mutations cluster in the N-lobe and presumably promote dimerization and independence of cytokine-receptor complex formation to become active and maintain activated JAK-STAT signaling. It is tempting to speculate that inhibitors which selectively target mutations in this region could block variant-driven dimerization while leaving a wild-type JAK protein unaffected and responsive to physiological signals. Likewise, clustering of FERM domain GoF mutations distal from cytokine receptor interactions is observable in the overlay of JAK1–JAK3. Further research is needed to address these areas for allosteric inhibitor targeting.

## Figures and Tables

**Figure 1 ijms-24-14573-f001:**
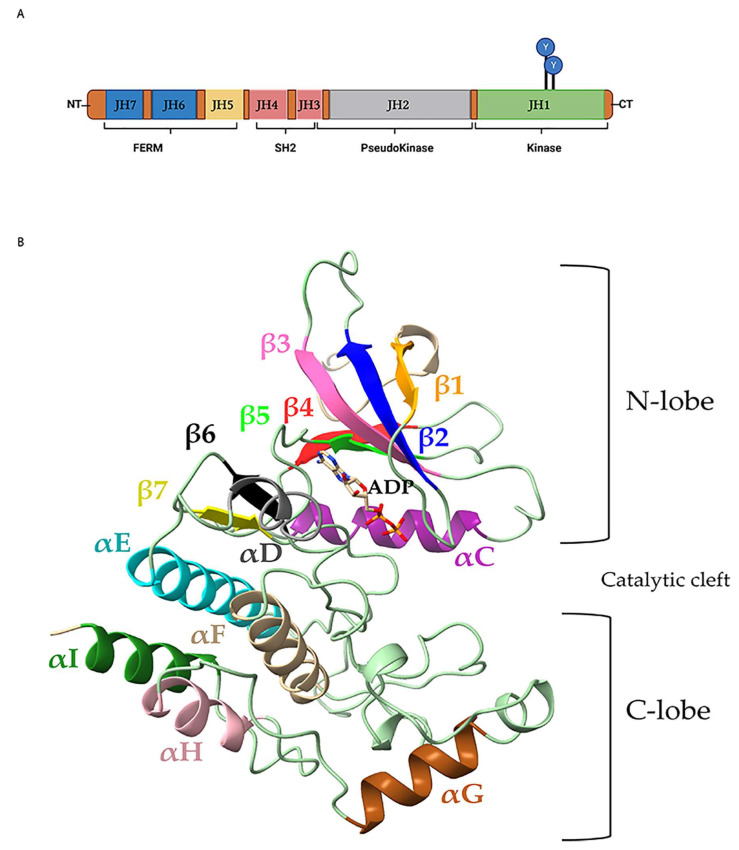
Linear structure of seven Janus homology (JH) domains and four functional domains of JAK family members. (**A**) JAK amino-terminal (NT) Janus homology 7 (JH7) domain and JH6 domain form the band 4.1, ezrin, radixin, moesin (FERM) segment; followed by a Src-homology 2 (SH2) consisting of the JH4 and JH3 domains; the JH2 pseudokinase domain and carboxyl-terminal (CT) JH1 kinase domain. JH1 activation loop regulatory tyrosine residues are depicted as Y. Created with BioRender.com. (**B**) Representative of JAK1 JH1 and JH2 ribbon structure depicting secondary α-helices and β-sheets within the N-lobe and C-lobes.

**Figure 2 ijms-24-14573-f002:**
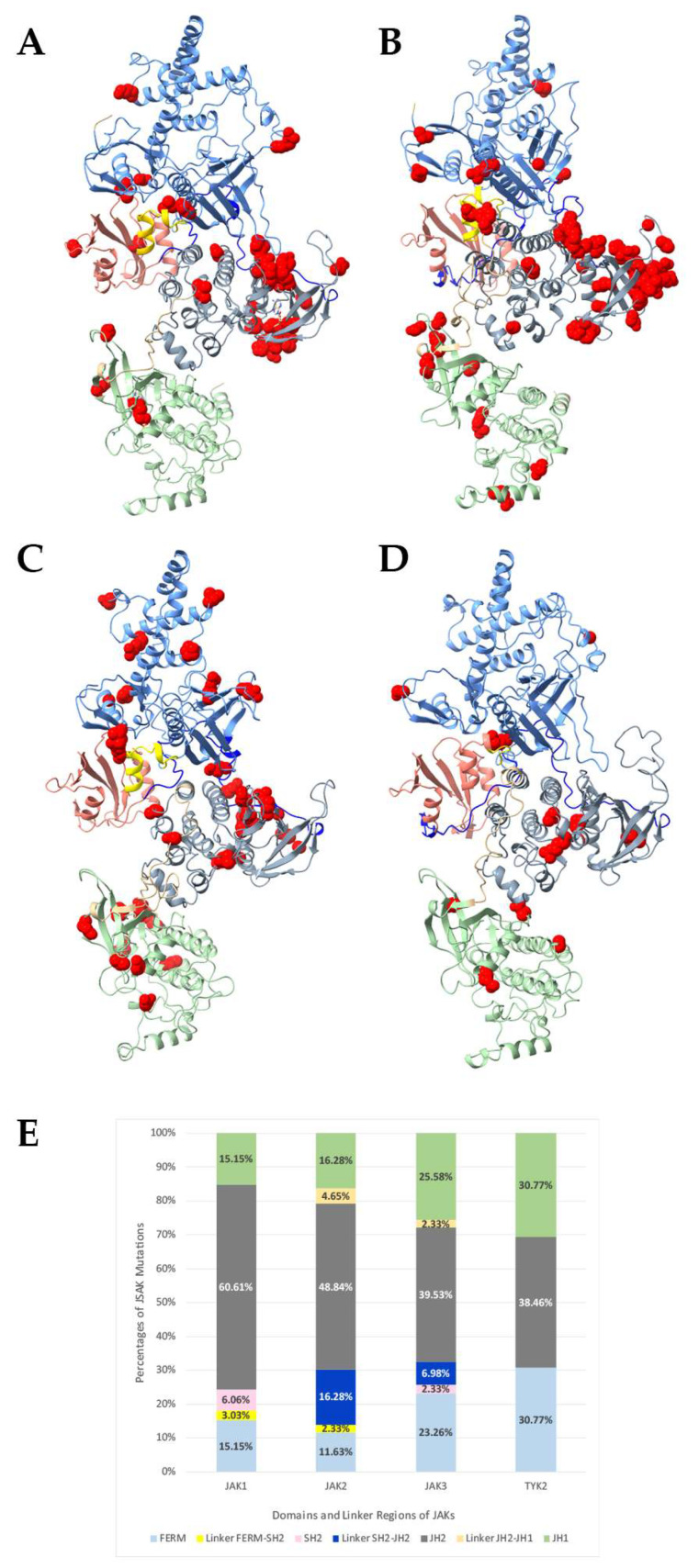
Mutational map of hematopoietic malignancy driving JAK variants. FERM domain (light blue), FERM-SH2 linker (yellow), SH2 domain (pink), SH2-JH2 linker (dark blue), pseudokinase domain (grey), JH2-JH1 linker (beige), and kinase domain (light green). The side chain of the residues associated with hematological mutations are depicted as red spheres. Human JAK1 (**A**), JAK2 (**B**), JAK3 (**C**), and TYK2 (**D**). (**E**) Bar-graph representing percentage of hematopoietic malignancy JAK mutations (from Table 1) within individual functional domains and linker regions (FERM-SH2 linker, SH2-JH2 linker, and JH2-JH1 linker combined).

**Figure 3 ijms-24-14573-f003:**
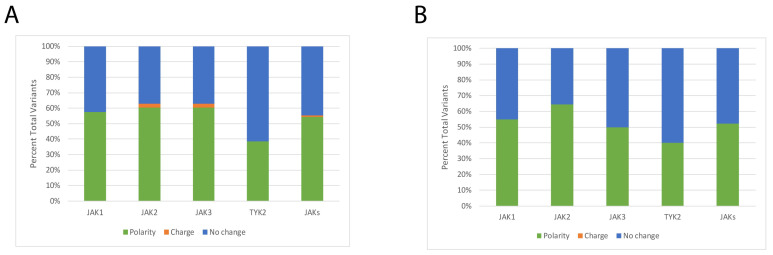
Nature of JAK amino acid variant change. (**A**) The nature of change for driver mutations was quantified across the full polypeptide sequence or (**B**) the pseudokinase domain alone across JAK family members. The bar labeled JAKs represents the compiled figures as a family. Change of polarity (green), charge (orange), and no change (blue) of individual residues were considered. Graphic based on macros-supported data analysis of compiled mutations from Table 1. Amino acids G, A, V, L, I, P, M, F, Y, and W were classified as non-polar; S, T, C, N, and Q as polar; K, H, and R as positively charged; and D and E as negatively charged.

**Figure 4 ijms-24-14573-f004:**
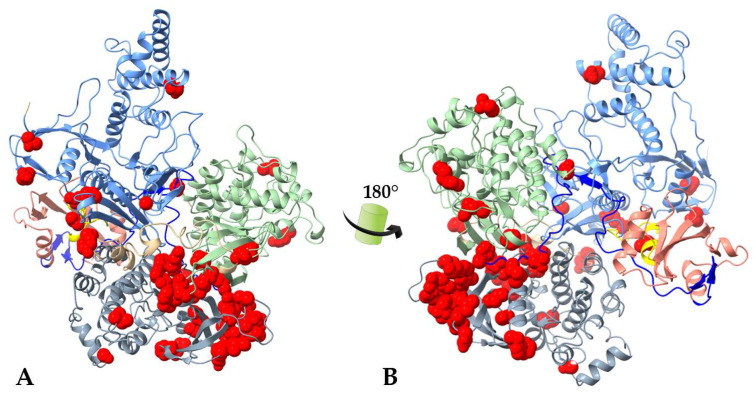
JAK2 closed structure GoF mutations associated with hemopoietic malignancies. Inactive closed-structure of JAK2 FERM domain (light blue), FERM-SH2 linker (yellow), SH2 domain (pink), SH2-JH2 linker (dark blue), JH2 domain (gray), JH2-JH1 linker (beige), and JH1 domain (light green). (**A**) Side chain of residues-reported mutations as drivers of enzymatic hyperactivity are shown in red spheres. JAK2 hyperactivating FERM mutations: E61K, T108A, E177V, G276A, R340Q, L393V; SH2 mutations: T514M, N533I/Y, M535I, K539L, I540T, 538–547 indels; pseudokinase mutations: F556L, R564Q, V567A, H587N, S591L, H606Q, K607N, H608Y, L611S, V617F, C618R, D620E, L624P, I645V, I682F, R683S/G, S755R, Y813D, E846D; kinase mutations: R867Q, D873N, T875N, P933R, R938Q, R1063H, N1108S are shown. (**B**) Closed structure was rotated 180° along the vertical axis to show the back side of JAK2.

**Figure 5 ijms-24-14573-f005:**
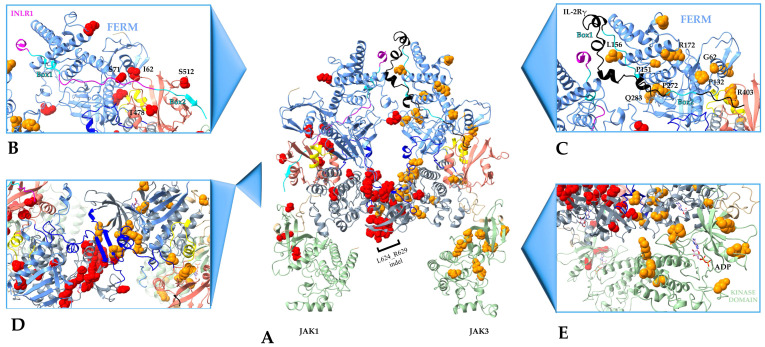
Landscape of hematopoietic malignancy driving mutations in the JAK1-JAK3 dimer. (**A**) Dimer of JAK1–JAK3 (left–right, respectively). FERM domain is depicted in light blue, FERM-SH2 linker in yellow, SH2 domain in light pink, SH2-JH2 linker in royal blue, pseudokinase in gray, JH2-JH1 linker in beige, and kinase domains in green. JAK1 mutations are portrayed as red spheres and JAK3 mutations as orange spheres for sites reported in Table 1. JAK1 is bound to partial intracellular *IFNLR1* (purple). *IFNLR1* Box1 and Box2 motifs depicted in cyan. JAK3 is bound to the IL-2Rγ intracellular region (black). IL-2Rγ Box1 and Box2 motifs are in cyan. Residues composing the JAK1 L624_R629 indel are marked. (**B**) JAK1 FERM region (light blue) complexed with *IFNLR1* (purple) with Box1 and Box2 in cyan. (**C**) JAK3 FERM region (light blue) showing interaction with IL-2Rγ (black). (**D**) JAK1 kinase domain portraying mutations from Table 1 as red spheres the sites for mutations. Transversal cut in the region where JAKs interact showing the sites for mutation at this level. (**E**) JAK3 kinase domain (light green) portraying mutations from Table 1 as orange spheres around the active JH1 domain.

**Figure 6 ijms-24-14573-f006:**
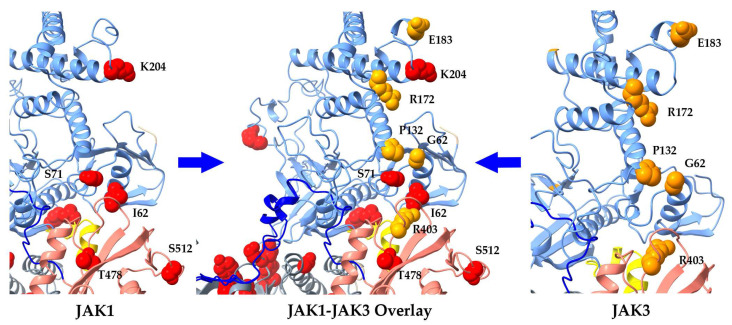
Overlay of JAK1 and JAK3 FERM-SH2 domains reveals hot spot for hematopoietic malignancy driving variants. JAK FERM domain (light blue), SH2 domain (pink), and FERM-SH2 linker (bright yellow) depict the location of JAK1 variants (**left**, red spheres), JAK3 variants (**right**, orange spheres), and an overlay of JAK1–JAK3 variants (**center**). JAK1 I62V (B-ALL; T-ALL), S71C (ETP-ALL), K204M (B-ALL), T478S (AML), and S512L (T-ALL) and JAK3 G62S (AML), P132A(AMKL), R172Q (ATLL), E183G (ATLL), R403H(T-ALL) variants are depicted.

**Figure 7 ijms-24-14573-f007:**
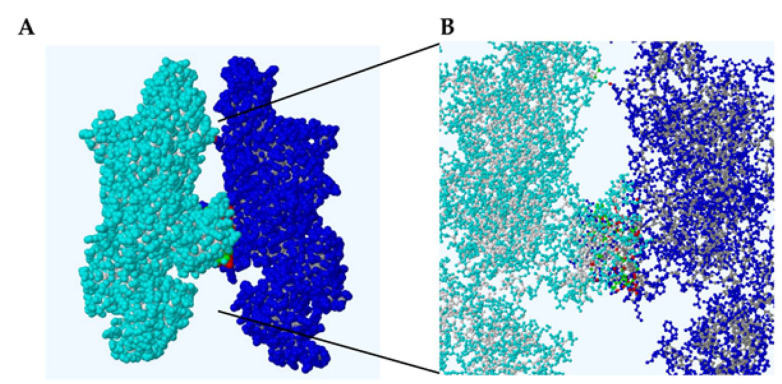
JAK1-JAK3 PDBe PISA Structure. Full-length wild-type JAK1 (cyan) with contact residues depicted in green and wild-type JAK3 (blue) with contact residues depicted in red. (**A**) Spacefill surface. (**B**) Ball-stick heterodimer structure.

**Table 1 ijms-24-14573-t001:** Partial list of Summarized JAK Hematopoietic Malignancy Variants. (For the full list, refer to Appendix A). Acute lymphoblastic leukemia (ALL); Early T-cell precursor (ETP); Acute myeloid leukemia (AML); Polycythemia vera (PV); Myeloproliferative neoplasm (MPN); Essential thrombocythaemia (ET); Acute megakaryoblastic leukemia (AMKL); Adult T-cell leukemia/lymphoma (ATLL); T-cell prolymphocytic leukemia (T-PLL); Juvenile myelomonocytic leukemia (JMML); Natural killer/T-cell lymphoma (NKTCL); Idiopathic myelofibrosis (IMF); Down syndrome ALL (DS-ALL).

JAK	Region	Mutation	Hematopoietic Malignancy Subtype
JAK1	FERM	I62V	B-ALL; T-ALL
		S71C	ETP-ALL
		K204M	B-ALL
	SH2	T478S	AML
		S512L	T-ALL
	JH2	V658F	ALL
		S703I	ALL
		R724H	B-ALL; T-ALL
		R724Q	B-ALL; T-ALL
		R724S	B-ALL; T-ALL
	JH1	R879S	T-ALL
		R879C	T-ALL
		R879H	T-ALL
		T901G	T-ALL
		K908T	Pediatric T-ALL
		K908A	Inactive JAK1
JAK2	FERM	E61K	Putative primary erythrocytosis
		T108A	Found as germline mutation in V617F-positive PV patient
		E177V	Putative primary erythrocytosis
		G276A	Putative primary erythrocytosis
		R340Q	PV
	FERM-SH2 Linker	L393V	Found as germline mutation in V617F-positive PV patient
	SH2-JH2 Linker	T514M	MPNs
		N533I	PV (together with K539L)
		N533Y	PV (together with K539L)
		M535I	AMKL
		K539L	MPNs
		I540T	PV
		538–547 indels	MPNs
	JH2	F556L	MPNs
		R564Q	Hereditary ET
		V567A	MPNs
		H587N	MPNs
		S591L	MPNs
		H606Q	MPNs
		K607N	AML
		H608Y	MPNs
		L611S	ALL
		V617F	MPNs
		C618R	MPNs
		D620E	MPNs
		L624P	MPNs
		I645V	MPNs
		I682F	ALL
		R683G	MPNs
		R683S	MPNs
		S755R	Hereditary thrombocythemia (together with R938Q in cis)
	JH2-JH1 Linker	Y813D	IMF
		E846D	Germline mutation found in erythrocytosis and megakaryocytic atypia
	JH1	R867Q	ALL, hereditary thrombocythemia
		D873N	ALL
		T875N	AMKL
		P933R	ALL
		R938Q	Hereditary thrombocythemia (together with S755R in cis)
		R1063H	Germline mutation found in erythrocytosis and megakaryocytic atypia
		N1108S	PV
JAK3	FERM	G62S	AML
		P132A	AMKL
		R172Q	ATLL
		E183G	ATLL
	SH2	R403H	T-ALL
	JH2	Q507P	T-PLL
		M511I	T-PLL, AML, JMML, NKTCL
		A573V	DS-ALL, DS AMKL, NTCL
		A593V	Transform BaF3 cells
		R657Q	T-PLL, AML, JMML, NKTCL
		V765D	JMML

**Table 2 ijms-24-14573-t002:** Summary of wild-type JAK heterodimer PDBe PISA analysis. Predicted accessible surfaces, dimer interface, salt bridge, and hydrogen bond formation between JAK1-JAK3 heterodimer.

Monomer	JAK1		JAK3	
**Class**	Protein		Protein	
**Number of residues**				
Interface	24	2.10%	17	1.60%
Surface	1111	98.90%	1045	98.70%
total	1123	100.00%	1059	100.00%
**Solvent-accessible area, Å**				
Interface	751.7	1.20%	897.5	1.50%
Total	61,180.1	100.00%	58,264.5	100.00%
**Solvation energy, kcal/mol**				
Isolated structure	−1019.9	100.00%	−942	100.00%
Gain on complex formation	−4.2	0.40%	−5.6	0.60%
Average gain	−1.5	0.001	−2.5	0.003
*p*-value	0.166		0.13	
**Interface Salt bridges**				
	**JAK1**	**Dist. [Å]**	**JAK3**	
1	GLU 637	2.00	LYS 515	
2	GLU 637	3.21	LYS 515	
3	ARG 577	2.93	GLU 571	
**Interface Hydrogen Bonds**				
	**JAK1**	**Dist. [Å]**	**JAK3**	
1	PHE 575	3.15	MET 511	
2	LEU 573	3.36	PHE 513	
3	SER 571	3.17	LYS 515	
4	ARG 577	3.64	LEU 508	
5	ARG 577	3.20	SER 509	
6	ARG 577	3.77	SER 509	
7	PHE 575	3.39	MET 511	
8	LEU 573	3.18	PHE 513	
9	ARG 659	3.56	GLU 567	
10	ARG 577	2.93	GLU 571

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
