# Peer review of "Structural Analysis of Janus Tyrosine Kinase Variants in Hematological Malignancies: Implications for Drug Development and Opportunities for Novel Therapeutic Strategies"

_ijms, 2023, doi:10.3390/ijms241914573_

Round 1

Reviewer 1 Report

See attached file

Minor editing of English language required

Reviewer 2 Report

In this paper, the authors performed molecular modelling of the open forms of the JAK1, JAK2, JAK3 and TYK2 proteins, the closed form of the JAK2 and the JAK1-JAK3 dimer. By exploiting these structures, they analyzed the locations and structural relationships between known mutational sites of the proteins.

The paper is well-written and the obtained results and interesting and overall clearly explained. I have just some suggestions:

1.      It would be better if the authors used the gene name IFNLR1 for Interferon lambda receptor 1, instead of INFR1 (https://www.uniprot.org/uniprotkb/Q8CGK5/entry)

2.      The correct name of the modelling server is “Swiss-Model” and not “Swiss-Modeling”

3.      The meaning of columns “NatOR AA*” and “New Nat*” in Table 1 is not clear and should be specified in the legend

4.      Given its length, I would move Table 1 to Supplementary Materials

5.      In section 2.2, the authors should better explain in the text why they selected the JAK2 protein as an example of a closed structure

6.      If I correctly understood, the authors used the PDBID: 4OLI as a template to model the JAK2 closed structure. Then, the citation for the PDB structure 4OLI is incorrect (line: 277) and should be changed to: https://doi.org/10.1073/pnas.1401180111 (from https://www.rcsb.org/structure/4OLI). Citations for the PDB structures should also be added in the methods close to their PDB entries.

7.      In the methods, the authors should specify which sequences they used as reference for each of the four proteins modelled

8.      References for SWISS-MODEL and ChimeraX are missing

Round 2

Reviewer 1 Report

I have appreciated the authors’ efforts to appropriately address all the points raised and for having made the necessary changes.

I am fully satisfied with the additions to the manuscript and can recommend publication.